# Quantifying Emergence in Neural Networks: Insights from Pruning and Training Dynamics

## Abstract

Emergence, where complex behaviors develop from the interactions of simpler components within a network, plays a crucial role in enhancing neural network capabilities. We introduce a quantitative framework to measure Emergence as the structural nonlinearity in networks, study the dynamics of this measure during the training process, and examine its impact on network performance, particularly in relation to pruning and training dynamics. Our hypothesis posits that the degree of Emergence — defined by the distribution and connectivity of active nodes — can predict the development of emergent behaviors in the network. We demonstrate that higher Emergence correlates with improved training performance. We further explore the relationship between network complexity and the loss landscape, suggesting that higher Emergence indicates a greater concentration of local minima and a more rugged loss landscape. We show that this framework can be applied to explain the impact of pruning on the training dynamics. These findings provide new insights into the interplay between Emergence, complexity, and performance of neural networks, offering implications for designing and optimizing architectures.

## 1. Introduction

Emergence is a phenomenon where complex behaviors arise from the interactions of simpler elements within the network. Understanding and leveraging Emergence is critical for further enhancing the capabilities of artificial neural networks. However, a notable lack of work has been on defining Emergence and its practical applications for improving network architectures (Li et al., 2023a). Emergence in neural networks encapsulates the dynamics of information flow, where intricate patterns and functionalities arise from the interaction between active and inactive nodes. This emergent behavior is not explicitly programmed but emerges through the network's ability to propagate and integrate information during the training process, leading to the network's ability to solve complex tasks. This paper delves into the theoretical aspects of Emergence in neural networks and validates our findings through empirical studies.

Our primary objective is quantifying Emergence during neural network training and exploring its correlation with performance. Understanding how Emergence evolves can help us identify when a network begins to specialize and focus on relevant features. By tracking this process, we can predict when a model is nearing its optimal performance, allowing for earlier intervention or fine-tuning. Evaluating a network's structures and weight values using our method provides valuable insights into its potential performance during training. This approach can guide optimization, reducing training time and computational resources.

We measure Emergence as the structural nonlinearity in the neural network as it grows in scale, hypothesizing that such quantity is a predictor of the development of emergent traits (Han et al., 2015). The results of our experiments indicate a correlation between higher levels of Emergence and improved training performance of the networks (Ruder, 2016). Networks exhibiting greater Emergence tend to converge more efficiently and achieve higher accuracy. Additionally, we explore the concept of network complexity and its spatial representation within the loss landscape. Our observations indicate that Emergence might reflect the concentration of potential local minima, implying that networks with higher Emergence could navigate the loss landscape more effectively.

Pruning is also examined in the context of Emergence. While pruning leads to faster convergence and enhanced training efficiency(Han et al., 2015), it typically results in a reduction of final accuracy. This trade-off highlights the importance of balancing the number of parameters and efficiency in the design of neural network architectures (Nair and Hinton, 2010). By eliminating the weights in the neural network and thus the activity of nodes, pruning affects our Emergence value. Good pruning algorithms that keep the network performance high also keep the Emergence value high; we validate

the idea that our quantitative framework to measure Emergence correlates with performance and training capabilities.

**Emergence Increases with scale:** Emergence ($E$) is inherently tied to the network's overall scale and connectivity, which increases with the number of parameters and layers. Higher Emergence values in larger, more complex networks indicate a stronger potential to develop sophisticated behaviors and patterns during training. Intuitively, complexity in this context refers to the network's capacity to represent intricate functions and patterns, often a function of its architecture and the number of parameters it possesses. As a measure of scale and connectivity, Emergence suggests that as networks become larger, they have a higher potential for exhibiting emergent traits, leading to more nuanced and sophisticated behaviors (Li et al., 2023a).

We have also defined relative Emergence, which investigates the Emergence of the model relative to the number of parameters in the model and gives us a way to find Emergence relative to the size of the network.

**Relative Emergence Correlates with training performance:** Relative Emergence ($\tilde{E}$) provides a metric for evaluating how efficiently a network's scale contributes to its learning process. Trainability refers to the ease and efficiency with which a neural network can learn from data and improve its performance during training. A higher relative Emergence in pruned networks suggests that these networks, although simpler, are more adept at learning and adapting, leading to faster convergence and improved performance during training. This suggests that some smaller models, despite having fewer parameters, are better at leveraging their complexity for effective learning (Ruder, 2016).

Additionally, we discuss the interpretation of Emergence from the loss function landscape perspective. Models with strong Emergence can be trained on various tasks, which suggests the existence of multiple local minima within a certain proximity to the network's current placement in the loss landscape. If the Emergence value is low, it could indicate a flat region of the loss function landscape. This insight allows us to understand the local geometry of the loss landscape and predict the network's training behavior.

Our work builds on and is inspired by recent theoretical advancements, such as the framework presented by (Li et al., 2023a). This work provides a rigorous mathematical foundation for understanding Emergence in network structures, which we extend and apply to the training dynamics of neural networks. By integrating these theoretical insights with empirical validation, our findings offer new perspectives on the role of Emergence in neural network performance and complexity. Our findings offer new insights into the theoretical underpinnings of Emergence and complexity in neural networks. We can develop more efficient and effective neural network architectures by understanding how Emergence influences network performance and complexity.

## 2. RELATED WORK

The concept of Emergence has been explored in various contexts within machine learning. In large language models, Emergence refers to the phenomenon where new skills appear as the model's scale, in terms of parameters and training data, increases, which can be linked to the model's ability to perform tasks it was not explicitly trained for, such as in-context learning or zero-shot learning (Arora and Goyal, 2023). In a similar framework, (Wei et al., 2022) defines Emergence in large language models as the phenomenon where certain abilities are absent in smaller language models but appear in larger ones, making these abilities unpredictable based on smaller-scale models. Another perspective on Emergence focuses on emergent abilities through the lens of pre-training loss rather than model size or training compute, as discussed by (Du et al., 2024).

Emergence has also been explored in various contexts in network systems, similar to how we explore Emergence in neural networks. (Siyari et al., 2019) classifies Emergence as the development of hierarchical modularity within complex systems, where smaller, function-specific modules combine to form larger, complex structures. This hierarchical structure is seen as a product of evolutionary processes that optimize the system for efficiency and robustness, often resulting in an "hourglass architecture" where the system of interest produces many outputs from many inputs through a relatively small number of highly central intermediate modules, these could be classified as the alive nodes. Emergence is also seen in a similar framework in (O'Brien et al., 2021), which defines Emergence within the context of network structures, particularly focusing on the Emergence of leader nodes in non-normal networks; this gives room for some of the nodes to be dead by our definition while the leader nodes are alive. Here, Emergence is tied to the directedness and asymmetry of the network, whereas the network becomes more non-normal, and leader nodes (nodes with no out-degree) spontaneously emerge, driving the dynamics and hierarchical organization within the network.

Emergence can also be defined in a mathematical sense, particularly concerning networks. (Li et al., 2023a) provides a categorical framework for quantifying emergent effects in complex systems through the lens of network topology. Their framework introduces a computational measure of Emergence that ties the phenomenon to the network's topology and local structures. This approach offers a novel method for quantifying Emergence, which could be applied to various systems, including machine learning models and biological networks. Emergence is also explored by (Li et al., 2024), who proposes a new neural network initialization scheme that enhances Emergence, which refers to the complex behaviors arising from simpler system components by adjusting layer-wise weight scaling, resulting in improved training speed and accuracy. This method demonstrates significant performance gains across various architectures.

There are a few machine learning aspects that are related to Emergence, such as the robustness and generalizability of neural networks. (Lei et al., 2019) looks at the relationship between the size of training data and the resulting accuracy and robustness of neural networks. Robustness refers to the ability of a neural network to maintain its performance in the presence of small, imperceptible changes to the input data that can cause a well-trained model to make incorrect predictions. As networks become more specialized through training, they might lose some generalization capability (or overall robustness), similar to the decline or stabilization of robustness. (Mangal et al., 2019), explore probabilistic robustness, which accounts for real-world input distributions and provides a practical approach for verification. Robustness ensures that the network performs reliably, and studying Emergence explores the internal dynamics that contribute to this performance.

## 3. METHODOLOGY

### 3.1. AN OVERVIEW OF THE MATHEMATICAL FRAMEWORK FOR EMERGENCE

Emergence fundamentally arises from the observation of a system from a higher scale. We build our definition of Emergence on the notion of nonlinearity as the information is passed to higher scales. Two key conceptual components are necessary to qualitatively describe emergent effects within the framework proposed by (Adam, 2017). The first is a notion of interaction or local computation among the components of a system. For example, the communication and propagation of information among nodes or subnetworks in the neural network. The second is the notion of interactional effects, which equips each system with an observable, for example, attaches network with its performance or abilities. These kinds of interactional effects are almost always associated with partial observations or a simplification and integration of lower — more foundational or granular —levels or scales in the system that result in a 'loss of information' or pattern/ feature formation at a higher level.

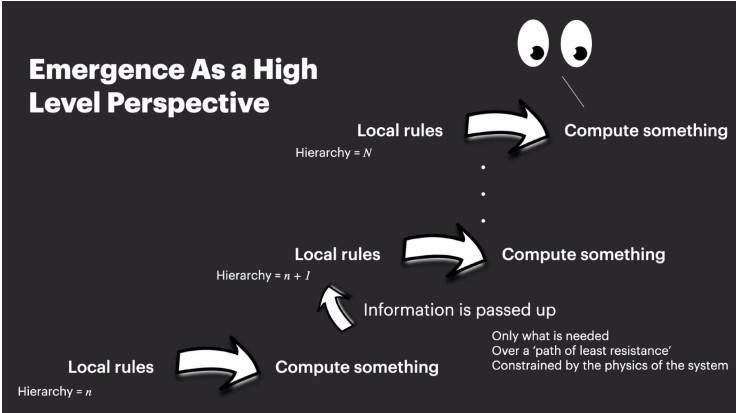

Figure 1: An illustration of Emergence in the hierarchical system.

With these two ingredients, we can define Emergence as a partial observation of interacting and interconnected components within a system that cannot be explained by known interactions that produce or result in partial observations of the components. This notion agrees with the intuitive understanding of Emergence that some properties of the interconnected components cannot be decomposed or reduced to combinations of known properties of the constituent components, i.e., that the whole is more than the sum of its parts. This notion of Emergence is the foundation on which our work in this paper, building on the framework first proposed in (Adam, 2017), develops a mathematical definition and computational measure of Emergence.

To formalize these ideas, we begin by representing the interactions between components as an operation $\vee$, where $s_1 \vee s_2$ represents a new interconnected system of subsystems $s_1$ and $s_2$. Interactional effects are described by the mapping $\Phi$ that sends a system to its partial observation or interactional effect at a higher scale, sometimes corresponding to a coarse graining scheme (Rosas et al., 2024). Emergent effects are sustained whenever the observation of the separate components cannot explain the observation of the combined system. Mathematically,

$$\Phi(s_1 \vee s_2) \neq \Phi(s_1) \vee \Phi(s_2)$$

for some constituent subsystems $s_1$ and $s_2$. Here, $s_1$ and $s_2$ are components of the system, $\Phi$ represents the observation or computational process, and $\vee$ is the binary operation encoding the interactions among the components. In machine learning context, we consider $s_1$ and $s_2$ as components of a model, and $s_1 \vee s_2$ represents the combined components. Emergent phenomena occur when the combined model's effect differs from the separate models' sum. Our study focuses on a single component's potential to exhibit emergent effects. (Li et al., 2023a) proposed a network-based measure of Emergence as:

$$\text{Emergence}(G, H) = \sum_{x \in G \setminus H} \#\text{paths in } H \text{ from } N_H(x) \text{ to } H,$$

where $G$ is the network at the lower scale, $H$ is the image of $G$ under the mapping $\Phi$, which can be considered as the representation of the network at the higher scale:

$$G \xrightarrow{\text{Cross-hierarchy mapping } \Phi} H$$

, and $N_H(x)$ represents the set of neighbors of node $x$ in network $H$. This paper represents the initial network as $G$ and the trained network as $H$, so $\Phi$ represents the training process.

### 3.1.1. MEASURING EMERGENCE IN NEURAL NETWORKS

In a machine learning setting, one modeling approach is to consider $\Phi$ as the training process; since Emergence here evaluates the potential/ ability for emergent traits when we observe system $G$ from a higher level $H$, here we want $G$ to represent the model itself, and $H$ to be some specific features of the model. In the paper, we adopt the setting that $H$ is the nodes in $G$ that are active in the training process, where active nodes are defined as the set of nodes whose activation is greater than a threshold. This sorted out the nodes that were not actively participating in the computational process. "The set of active nodes represents the pathways through which information flows, reflecting the network's ability to learn and adapt. The Emergence measure attempts to quantify this information flow by observing how active and inactive nodes contribute to the propagation and transformation of features. This aligns with our framework, where information is selectively integrated or neglected during the learning process. This fits in our framework of Emergence, where part of the system is being neglected after the learning process, thus the learning process represents the $\Phi$ where partial observation is carried out, and the properties of $H$ represents the emergent abilities of the network.

For MLP, consider its fully connected structure, the paths-counting is reduced to the following measure of Emergence (Li et al., 2024):

$$E = \sum_{i=1}^{N-1} \sum_{j>i}^{N} (n_i - a_i) a_j \prod_{k=i+1}^{j-1} n_k$$

where $N$ is the number of layers, $n_i$ is the total number of nodes in layer $i$, $a_i$ is the number of active nodes in layer $i$. For convolutional networks, the information flow is constrained by the pooling layers, we have:

$$E = \sum_{i=1}^{N-1} \sum_{j>i}^{N} (n_i - a_i) a_j \prod_{k=i+1}^{j-1} m_k$$

where $m_k$ is the number of filters in layer $k$.

For other architectures, the equation to compute Emergence will need to be slightly modified based on their specific structures. When the network architecture is fixed, meaning the number of layers

L and the number of nodes $n_i$ for each layer $i = 1, \ldots, L$, along with the number of filters $m_k$ for all pooling layers, are constant, Emergence becomes solely a function of the number of active nodes in each layer. Thus, Emergence is expressed as a polynomial $E(a_1, \ldots, a_N)$, where $a_1, \ldots, a_N$ represent the active nodes in each of the network's layers.

Emergence in neural networks is quantified by counting the number of paths from inactive nodes to active nodes. The network's weights influence the activity of nodes during initialization. By applying a criterion for active nodes, such as considering nodes whose activations exceed a given threshold (as adopted in this paper), we can define an activation-based measure of Emergence in neural networks.

This measure provides a way to observe the network's complexity and its potential for effective information flow. By analyzing the weight distribution and the activity of the nodes over time, we can infer the network's ability for learning and adapting to new patterns. To monitor the activation of a node, we track its output after applying the activation function. We log the activation levels for all nodes across various layers during each epoch. This data is then analyzed to assess the proportion of active versus inactive nodes, offering insights into the network's learning behavior. (Glorot et al., 2011). Over the course of training, a general trend of decreasing Emergence's value is observed as the network refines its feature representations and improves accuracy. This suggests that the network becomes more specialized, focusing on the most relevant features.

### 3.2. PREDICTING EMERGENCE IN THE TRAINING DYNAMICS

Emergence has been widely studied in neural networks as the network grows and new properties and abilities emerge. The natural question arises: What scale and structure would allow the network to exhibit these emergent properties? To answer this question, it is important to approximate the network size sufficient to train on a dataset. We start by building the measure of Emergence, which correlates with the model's ability to train on a given dataset. The Emergence measure evaluates the pathways of information flow between active and inactive nodes, as reflected in node activations and weight changes. By analyzing these pathways during early training, we capture the network's ability to propagate information effectively, which underpins its capacity to form relevant features and improve performance. Specifically, we calculate Emergence measures based on the distribution of active nodes. The dichotomy classification of the active and inactive nodes is motivated by exploring the feature formation inside the network, where nodes with higher activations represent certain features (Simonyan et al., 2013; Bau et al., 2017; Yosinski et al., 2015). As the number of paths is made by the weighted connection of the neural networks, from the inactive nodes to the active nodes, it essentially captures the potential paths of information flow that form feature representation in the network. As we show above, the pattern of information flow is related to model's nonlinearity in its response to input, this forms an evaluation of a model's ability to produce emergent traits.

### 3.3. EMERGENCE, RELATIVE EMERGENCE AND TRAINING DYNAMICS

The impact of Emergence on the training dynamics and loss-function landscape can be understood as follows. In the previous sections, we quantify Emergence as the number of paths from the inactive nodes to the active nodes; it essentially captures the potential paths of information flow that form feature representation in the network. If the network has a higher value of Emergence, there are more paths between base-nodes (inactive) to the feature representing nodes (active); the features are formed by summarizing over a larger amount of inputs. When networks exhibit such characteristics, they are more likely to form features (Shwartz-Ziv, 2022), corresponding to more local minimums (Li et al., 2018).

While absolute Emergence measures the overall scale and structure of a neural network, relative Emergence offers a normalized metric that accounts for the network's size. This normalization of Emergence elucidates how our theory correlates with network pruning and trainability.

In our study, we observe that pruned networks exhibit smaller absolute Emergence, reflecting their reduced number of parameters. However, when we normalize Emergence by the number of parameters in the pre-pruned network, we define this normalized metric as relative Emergence, denoted as $\tilde{E}$, providing deeper insights into the network's trainability. This relative Emergence is calculated as

$$\tilde{E} = \frac{E}{\# \text{ parameters}}, \tag{1}$$

where $E$ is the absolute Emergence, and the denominator represents the number of parameters in the network.

This relative Emergence, $\tilde{E}$, tends to be larger in pruned networks compared to non-pruned ones. This increase in relative Emergence suggests that, despite the reduced number of parameters and potential information pathways, pruned networks maintain a high degree of effective connections per parameter, which correlates with stronger trainability. In other words, properly pruned networks with fewer parameters are more efficient in developing emergent traits that improve their learning and generalization capabilities.

The implications of this relationship are twofold:

- **Emergence Increases with Scale**: Absolute Emergence ($E$) is inherently tied to the network's overall number of parameters and layers. Higher Emergence values in larger, larger networks with connectivity that allow optimal information flow networks indicate a greater likelihood of developing sophisticated behaviors and patterns during training.

- **Relative Emergence Correlates with Training performance**: Relative Emergence ($\tilde{E}$) provides a metric for evaluating how efficiently a network's limited parameters contribute to its learning process. A higher relative Emergence in pruned networks suggests that these networks, although simpler, are more adept at learning and adapting, leading to faster convergence and improved performance during training.

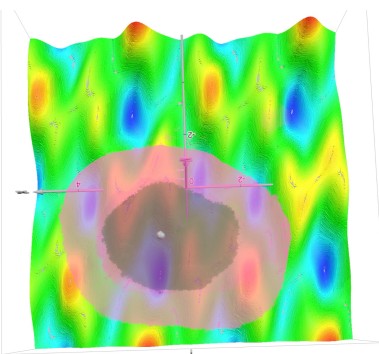

Figure 2: Visualization of Emergence in the loss landscape

### 3.3.1. INTERPRETING EMERGENCE FROM THE ENERGY LANDSCAPE PERSPECTIVE

Relative Emergence can be conceptualized as reflecting the density of local minima within a given region of the loss landscape. Although pruning reduces the overall 'size' of the region by decreasing the number of parameters, it effectively increases the density of local minima within this smaller region. This higher density of local minima implies that the pruned network, despite its reduced scale, possesses a more intricate and rich structure of optimization paths, facilitating faster and more efficient training convergence.

In Figure 2, we explore how one would visualize Emergence in the loss landscape pre- and post-pruning. The lighter pink region shows how the larger pre-pruned network covers a larger region than the darker region. The blue region represents the local minima. This distinction is critical because it highlights that while absolute Emergence is essential for developing complex behaviors, the efficiency of this network (relative Emergence) is crucial for practical trainability. By pruning the network, we reduce unnecessary parameters, thereby enhancing the network's ability to focus on only the most relevant features and pathways, which accelerates learning and convergence.

In the context of our methodology, this understanding aligns with our findings on the spatial representation of network complexity within the loss landscape. Emergence reflects the size of the region around the network's current state, indicating potential local minima. Pruned networks, with their higher relative Emergence, navigate this landscape more effectively, finding optimal solutions with greater efficiency.

Overall, the concept of relative Emergence not only corroborates our theoretical framework but also provides actionable insights for optimizing neural network architectures, balancing the number of parameters with trainability to achieve more efficient and effective learning models. We hypothesize that the quantity of Emergence, represented by the number of paths between active and inactive nodes, is indicative of the potential for emergent traits to arise later in the training process.

Furthermore, we propose that pruning the network will decrease the quantity of Emergence due to its reduced scale. This reduction in Emergence is expected to correlate with faster convergence in training and improvements in training efficiency and accuracy.

## 4. Experiments, Results, and Discussion

In our experiments, we conducted a comprehensive analysis of Multi-Layer Perceptrons (MLP) and Convolutional Neural Networks (CNN) to assess the impact of pruning on Emergence and training dynamics. We trained MNIST and Fashion-MNIST datasets on MLP, and we trained the CIFAR-10 dataset on VGG19, a convolutional architecture.

### 4.1. Experiments Setup

For the experiments reflected in Figures 2 - 7, the models were trained for 5 epochs on both the MNIST and Fashion-MNIST datasets to establish a baseline performance, achieving initial accuracies of 90.4% and 82%, respectively. Following this baseline, we created four identical copies of the trained model. These models were then subjected to different conditions: the first model continued training without pruning, serving as the control, with final accuracies of 95.7% and 86.3%, respectively; the second, third, and fourth models were pruned by 30%, 50%, and 70%. The pruning was executed using magnitude-based pruning to reduce the network's complexity systematically. The learning rate across all models was consistently maintained at 0.005 to ensure uniform training conditions. Nodes were classified as active if their activation exceeded 0.05, with those below this threshold deemed inactive. Emergence was quantified by counting the number of paths between alive and dead nodes, which served as a proxy for the network's complexity and its capacity for developing emergent traits.

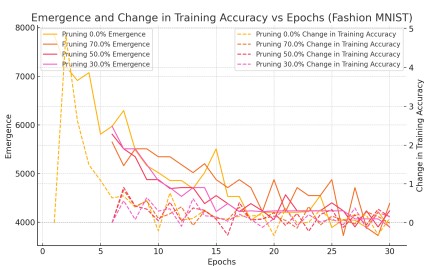

Figure 3: Emergence and Change in Training Accuracy vs Epochs ( Fashion-MNIST)

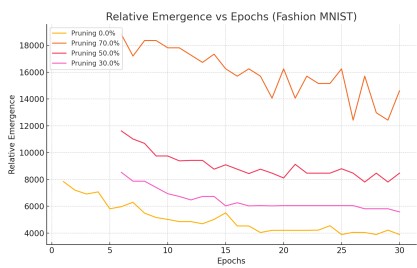

Figure 4: Relative Emergence vs Epochs ( Fashion-MNIST)

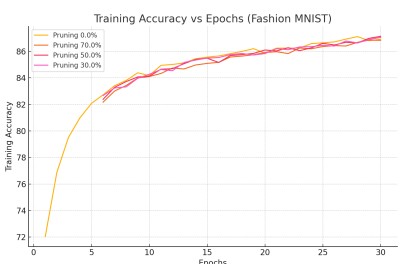

Figure 5: Training Accuracy vs Epochs ( Fashion-MNIST)

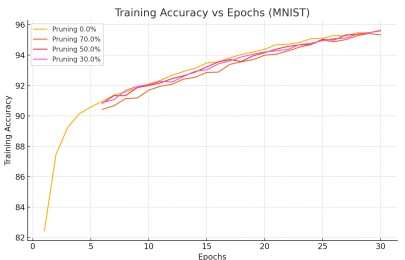

Figure 6: Training Accuracy vs Epochs (MNIST)

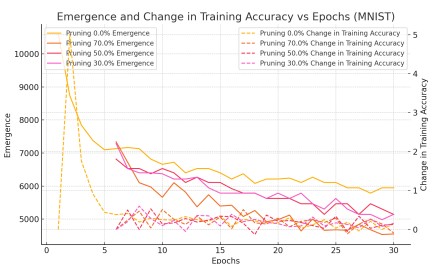

Figure 7: Emergence and Change in Training Accuracy vs Epochs (MNIST)

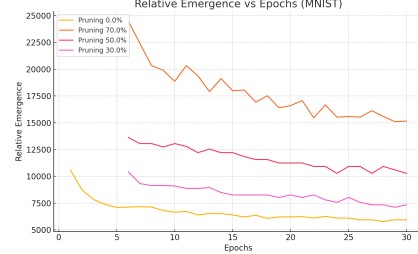

Figure 8: Relative Emergence vs Epochs (MNIST)

### 4.1.1. FASHION-MNIST RESULT ANALYSIS

The results obtained from the Fashion-MNIST dataset reveal a consistent trend: as the value of Emergence decreased, training accuracy change decreased, thereby supporting our hypothesis that Emergence functions as a measure of prediction of emergent traits within a neural network. Specifically, the control model, which underwent no pruning, reached a final accuracy of 86.3%, while the 30%, 50%, and 70% pruned models achieved final accuracies of 87%, 86.8%, and 86.2% respectively. The decrease in Emergence corresponded with increased network specialization, as evidenced by improved accuracy metrics. Pruning significantly reduced absolute Emergence due to the smaller network size; however, relative Emergence—normalized by the number of parameters—actually increased. This increase suggests that the pruned networks became more specialized, focusing on fewer but more relevant features. This enhanced specialization is reflected in the quicker convergence and improved training efficiency observed in the pruned models.

Figures 3 to 5 illustrate these dynamics, showing the relationship between Emergence, training accuracy, and the number of epochs. Notably, the 70% pruned model, despite its significantly reduced complexity, exhibited a remarkably high relative Emergence, indicating a more concentrated focus on essential learning pathways. This model achieved a faster convergence rate than its less pruned counterparts, albeit with a trade-off in final accuracy.

### 4.1.2. MNIST RESULT ANALYSIS

The MNIST dataset results exhibited the same trends observed in the Fashion-MNIST experiments, further validating our hypothesis. As Emergence decreased, training accuracy converged, with the control model achieving a final accuracy of 95.7%, and the pruned models achieving 95.7%, 95.6%, and 95.1%, respectively. The decrease in absolute Emergence was consistent across all pruned models, yet relative Emergence increased, particularly in the 50% and 70% pruned networks. This suggests that these pruned models, while simpler, had become more efficient in their learning processes, leveraging their remaining complexity more effectively. Figures 6 to 8 depict the same results as figures 3 to 5.

### 4.2. SIGNIFICANCE OF INITIALIZATION AND SCALE

To further explore the significance of initialization on network performance and Emergence, we conducted a series of experiments on different initialization of a wide range of network sizes. The first experiment was conducted under the same parameters as in Section 4.1. For the second experiment, the low Emergence models were created by altering the lower and upper bounds of the uniform distribution. For the small model, these bounds were set to [-0.01, 0.01], and the large model bounds were set to [-0.075, 0.075]. The medium Emergence models were created by altering the mean and standard deviation, setting the small model with 0 mean and 0.01 standard deviation, with the large model having 0 mean and 0.048 standard deviation.

### 4.2.1. PRE-TRAINING INITIALIZATION

In the first experiment, to achieve better initialization, we trained a network on the MNIST dataset for 25 epochs. To test the effect of scale, we split the model into two branches: one without pruning and one that was pruned.

Both branches were then further trained for another 25 epochs on a different but similar dataset, Fashion-MNIST. We also created two randomly initialized networks to test the effect of the initialization of the weights and, consequently, the Emergence activation value, one the size of the non-pruned network that was pre-trained on MNIST and one that is the size of the pruned network.

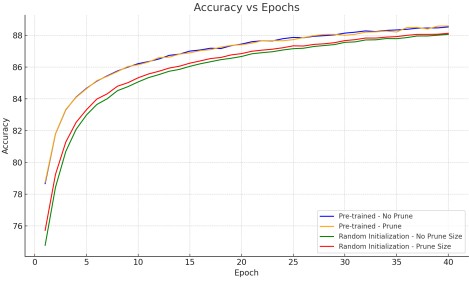
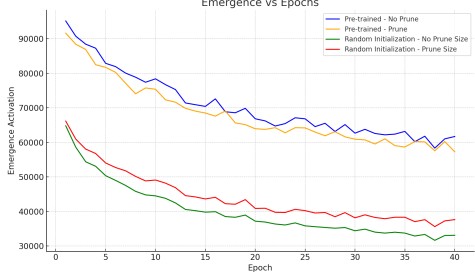

Figure 9: Training Accuracy vs Epochs          Figure 10: Emergence vs Epochs

The experiment showed how the pre-trained network on two different scales, pruned and non-pruned, have higher Emergence values than both the randomly initialized networks. Observing the experiment in the lens of the changing loss landscape, we can support the idea that high Emergence value relates to a better region to train a network. Since Fashion-MNIST is similar to MNIST, we expected the pre-trained networks to perform better. This better performance highlights how networks with initialization or starting points with higher Emergence value would ultimately achieve better training performance, supporting the idea that the Emergence value is a predictor of emergent traits.

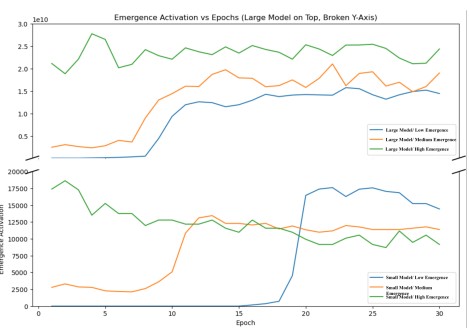

Figure 11: Emergence vs Epochs

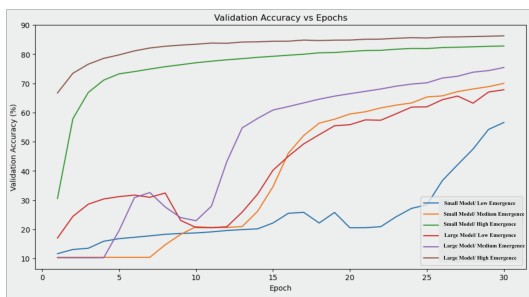

Figure 12: Training Accuracy vs Epochs

In the experiment depicted in Figures 11 and 12, we evaluated six distinct models, divided into two groups of three, each with varying initial Emergence values. The low Emergence and medium Emergence models were networks in which the weights were deliberately altered to yield suboptimal Emergence values, as opposed to the randomly initialized high Emergence models, which naturally yielded higher Emergence. The results demonstrate that all four manipulated models exhibited poor initial performance, as indicated by their low Emergence values and corroborated by their training accuracy. We hypothesize that these four models subsequently improved as the optimization function adjusted for the suboptimal initialization. This improvement is evidenced by the corresponding increases in both Emergence values and training accuracy during the experiment. The direct relationship between the change in training accuracy and the Emergence value validates the value being a predictor of the network performance.

### 4.3. CNN EXPERIMENTS WITH CIFAR-10

The experiments conducted on CNNs using CIFAR-10 dataset further validate our hypothesis. We observe a correlation between Emergence level and training performance (evaluated as the validation accuracy), depicted in Figures 13 and 14, which validates our hypothesis. The model with good Emergence takes the lead in the training performance.

On each trial base, similar to the observations in MLPs, we observed a convergence of Emergence in CNNs that aligned with the eventual convergence of accuracy. Emergence decreased significantly to zero in the later epochs as a consequence of all the nodes in the classifier layers becoming actively representing features. Zero Emergence suggests that no new emergent traits were forming, potentially making it a reliable indicator that the network will not experience significant improvements in performance in the future. When Emergence is low, accuracy improvements are minimal and appear random, indicating that the network is nearing its full potential. This highlights the importance of Emergence as a predictor of the network's learning capability and its potential for future performance gains.

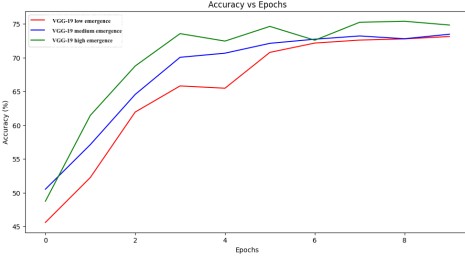

Figure 13: Emergence vs Epochs

Figure 14: Training Accuracy vs Epochs

### 4.4. DISCUSSION

Our study highlights the critical role of Emergence measure in understanding and optimizing neural network performance. Emergence, reflected as the connectivity and distribution of active nodes, serves as a predictor of the network's potential to develop complex, high-performing traits. Our results suggest that higher levels of Emergence may correlate with improved training performance. This observation raises the possibility that networks with greater complexity could be better

equipped to navigate the loss landscape effectively. Pruning significantly impacts both Emergence and performance. By reducing network complexity, pruning decreases absolute Emergence, leading to faster convergence. However, the relative Emergence—Emergence normalized by the network's size—increases in pruned networks, indicating a high concentration of local minima within their loss landscape. While pruned networks converge more quickly, they achieve lower final accuracy compared to non-pruned networks, which maintain higher levels of absolute Emergence and, ultimately, higher performance. This trade-off underscores the importance of balancing complexity and trainability in neural network design. The predictive capability of Emergence is another significant finding. As training progresses, a decrease in Emergence correlates with the network's approach to convergence. When Emergence drops to zero, further significant improvements in accuracy are unlikely, marking an optimal point to terminate training. This insight can optimize computational resources and streamline training processes. Understanding the role of Emergence allows for more informed decisions regarding network complexity and pruning strategies, ultimately contributing to the development of more efficient and effective neural network architectures.

Future research should focus on further validating these findings across different network architectures and datasets. Additionally, exploring the theoretical underpinnings of Emergence in more depth, including its mathematical modeling and spatial representation within the loss landscape, will provide a stronger foundation for applying these concepts in practical scenarios. Investigating the impact of various pruning techniques and their timing relative to the training process will also offer valuable insights into optimizing neural network performance.

## 5. CONCLUSION

In this paper, we have investigated the concept of Emergence in artificial neural networks, emphasizing its theoretical foundations and empirical validation. Our findings suggest that Emergence, defined by the connectivity between active and inactive nodes, could serve as a useful metric for understanding network performance. Our experiments validate that higher Emergence correlates with improved training performance. We also explored the implications of network complexity and its spatial representation within the loss landscape, revealing that higher Emergence indicates a more effective navigation of the loss landscape. Furthermore, we examined the effects of pruning on Emergence and network performance, showing that while pruning enhances training efficiency, it typically results in lower final accuracy. Our work, inspired by recent theoretical advancements, provides new insights into the role of Emergence in neural network performance, offering significant implications for the design and optimization of efficient neural network architectures.

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

## APPENDIX A. DERIVATION OF THE EMERGENCE MEASURE

To perform the mathematical computation of Emergence, we use quiver representation as the representation of a neural network. Formally, a **quiver** is a directed graph where loops and multiple arrows between two vertices are allowed, defined as follows:

- A quiver $Q$ is a quadruple $Q = (Q_0, Q_1, h, t)$ where $Q_0$ is a finite set of vertices, $Q_1$ is a finite set of arrows, and $h$ and $t$ are functions $Q_1 \rightarrow Q_0$. For an arrow $a \in Q_1$, $h(a)$ and $t(a)$ are called the head and tail of $a$.

- We get a **representation** $V$ of $Q = (Q_0, Q_1, h, t)$ if we attach to every vertex $x \in Q_0$ a finite dimensional vector space $V(x)$ and to every arrow $a \in Q_1$ a linear map $V(a) : V(ta) \rightarrow V(ha)$.

Quiver representation can be used to model the dynamics on the network (Derksen and Weyman, 2017; Armenta and Jodoin, 2021; Armenta et al., 2023). We provide two examples of quiver representation in Figure 14.

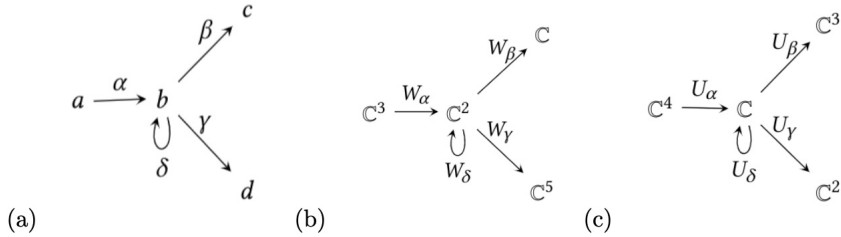

(a)                    (b)                         (c)

Figure 15: Additional examples of quivers. (a): A quiver $Q$ with vertices $V = \{a, b, c, d\}$ and oriented edges $E = \{\alpha, \beta, \gamma, \delta\}$, (b) and (c): two quiver representations over $Q$. Adapted from (Derksen and Weyman, 2017).

**Theorem** (Proposition 5.3 in (Li et al., 2023b)) Given the functor $\Phi$ which preserves partial structure in a quiver representation $W$ by deleting a set of edges $E$, the derived functor of $\Phi$ is

$$R^1\Phi(W) = \bigoplus_{a \in E} \Phi(W(ta) \otimes P_{ha}) \tag{2}$$

where $ta$ is the tail of edge $a$ (the starting node), $ha$ is the head of edge $a$ (the ending node), $W(ta)$ is the vector space associated to node $ta$, $P_{ha}$ is the vector space spanned by all paths originating from node $ha$.

**Proof**

Based on derksen2017introduction, for representation $W$ in $\mathbf{Rep}(Q)$ we have the projective resolution

$$0 \longleftarrow W \xleftarrow{f^W} \bigoplus_{x \in Q_0} W(x) \otimes P_x \xleftarrow{d^W} \bigoplus_{a \in Q_1} W(ta) \otimes P_{ha} \longleftarrow 0 \tag{3}$$

where

$$f^W : \bigoplus_{x \in Q_0} W(x) \otimes P_x \to W \tag{4}$$

is defined by

$$f^W(w \otimes p) = p \cdot w, \tag{5}$$

and

$$d^W : \bigoplus_{a \in Q_1} W(ta) \otimes P_{ha} \to \bigoplus_{x \in Q_0} W(x) \otimes P_x \tag{6}$$

is defined by

$$d^W(w \otimes p) = (a \cdot w) \otimes p - w \otimes pa. \tag{7}$$

Now we compute the first left derived functor $R^1\Phi$. By definition rotman2009introduction, it is the 1st homology object of the sequence above under the image $\Phi$, formally, $R^1\Phi = \ker \Phi d^W$, where $d^W$ is defined in (6). Now if an edge $a$ is deleted by the functor $\Phi$ then for any $w \in W(ta)$ and $p \in P_{ha}$, we have $(a \cdot w) \otimes p = w \otimes ap = 0$, hence $\Phi(W(ta) \otimes P_{ha}) \subseteq \ker \Phi d^W$. If $a$ is preserved under $\Phi$, then $\Phi d^W$ will act the same as $d^W$ on $\Phi(W(ta) \otimes P_{ha})$, and $d^W$ is injective due to the exactness of resolution, $\Phi(W(ta) \otimes P_{ha})$ will be non-zero thus not contribute to $\ker \Phi d^W$.

∎

This theorem computes $R^1\Phi$, which evaluates the difference between $\Phi(s_1 \vee s_2)$ and $\Phi(s_1) \vee \Phi(s_2)$, thus encodes the potential of a system for Emergence. Taking advantage of this theorem, we can take the dimension of $R^1\Phi(W)$ as a numerical approximation of the potential for Emergence of $W$ when the network interacts with other networks:

$$\dim R^1\Phi_l(W) = \dim \bigoplus_{e \in E} \Phi_l(W(he) \otimes I_{te})$$
$$= \sum_{e \in E} \dim \Phi_l(W(he)) \times \dim \Phi_l(I_{te}). \tag{8}$$

Here $\dim \Phi_r(V(te))$ and $\dim \Phi_l(W(he))$ is the dimension of the image of the vector space $V(te)$ and $W(he)$ under the functor, and $\dim \Phi_r(P_{he})$ and $\dim \Phi_l(I_{te})$ is the dimension of the image of the path algebra $P_{he}$ and $I_{te}$ under the functor.

Given a network $G$, and a sub-network $H$ which represents its effect or observation under $\Phi$, where their relation are shown as follows:

$$G \xrightarrow{\text{Cross-scale mapping } \Phi} H$$

then we have the following measure of Emergence for networks:

$$\text{Emergence}(G, H) = \sum_{x \in G \setminus H} \#\text{paths in } H \text{ from } N_H(x) \text{ to } H, \tag{9}$$

where $H$ represents the part of network structure being preserved by $\Phi$, the partial observation. $N_H(x)$ is the set of downstream neighbors of $x$ in $H$.

## APPENDIX B. EMERGENCE-PROMOTING INITIALIZATION SCHEMES

(Li et al., 2024) proposed the following initialization scheme, which has been shown to have higher value of Emergence:

We first initialize the network weights $\{W_i\}$ following some standard initialization scheme which preserves stability, for example, Xavier or Kaiming He Initialization. Then we do the following scaling to the weights:

$$\tilde{W}_{-n} = W_n/\alpha^n$$
$$\tilde{W}_{-(n-1)} = W_{-(n-1)}/\alpha^{n-1}$$
$$\vdots$$
$$\tilde{W}_0 = W_0$$
$$\tilde{W}_1 = W_1 * \alpha \tag{10}$$
$$\vdots$$
$$\tilde{W}_{n-1} = W_{n-1} * \alpha^{n-1}$$
$$\tilde{W}_n = W_n * \alpha^n$$

For example, when we initialize the VGG-19 model with different levels of Emergence, we adopted the following scheme:

We only consider the 3 fully connected layers at the end (the classifier). We have three layers of weights $W_1, W_2, W_3$, initialized by Kaiming Initialization. For model with good Emergence:

$$\tilde{W}_1 = W_1/2$$
$$\tilde{W}_2 = W_2 \tag{11}$$
$$\tilde{W}_3 = W_3 * 2.$$

For model with medium Emergence:

$$\tilde{W}_1 = W_1$$
$$\tilde{W}_2 = W_2 \tag{12}$$
$$\tilde{W}_3 = W_3.$$

For model with low Emergence:

$$\tilde{W}_1 = W_1 * 2$$
$$\tilde{W}_2 = W_2 \tag{13}$$
$$\tilde{W}_3 = W_3/2.$$