# OpenReview forum: "Quantifying Emergence in Neural Networks: Insights from Pruning and Training Dynamics"
_ICLR.cc/2025/Conference — Submitted to ICLR 2025_

### Official Review · Reviewer_wNmp · 2024-11-02

**Soundness:** 2
**Presentation:** 1
**Contribution:** 2
**Rating:** 1
**Confidence:** 4

**Summary:**

The authors suggest a measure of sparseness they term emergence as a key property of neural networks. The measure is the number of paths between active and inactive nodes over training. There are several experiments in which networks are trained and emergence is measured. In some of these experiments, there is a relationship between the two.

**Strengths:**

Understanding how representations form while a neural network is training is an important open problem. The authors suggest a new measure to this end.

**Weaknesses:**

Overall, this seems like an initial exploration that is not yet ready for publication.
Major points are:
1.	The paper uses many vague terms such as “intricate patterns”, “emergence” that are not properly defined.
2.	The actual definition of emergence is a form of sparseness. But the paper does not discuss any work on sparseness in machine learning. Perhaps LASSO is a good place to start.
3.	The main claim is a relation between emergence (or relative emergence) and performance. But there are no plots directly comparing the two.
4.	The experiments are without any repetitions (different seeds, etc.), and thus it is hard to draw conclusions.
5.	The figures are all without captions.
6.	The work is based on the framework of Li et al 2023, which is an arxiv preprint. Because it is an unreviewed preprint, the current paper should contain the essential elements rather than refer to the previous work.

**Questions:**

7.	Vague wording: ” Higher emergence values in larger, more complex networks indicate a stronger potential to develop sophisticated behaviors and patterns during training”
8.	REF Lei et al 2019 – no bibliographic information
9.	Definition of emergence is given without intuition. Why the set x\in G\H? Using H as a trained network suggests that nodes x do not disappear over training. The next paragraph defines active nodes, but the order can be confusing.
10.	Line 216 has the definition of emergence. It would be helpful to introduce this earlier, as it would help the reader to follow the equations.
11.	Line 234: suddenly emergence can also be measured by weight change.
12.	Figure 2: There in no caption to describe what exactly is plotted here. What do the colors mean? What do the shadings mean? Is this based on some simulations? Is this an hypothesis?
13.	Pruning is mentioned several times, but the exact process is not described.
14.	Results: change in accuracy is not defined (derivative over epochs?). Are there any statistics on accuracy? The differences in figure 5 between curves seem very noisy, and perhaps due to random seeds and not to pruning level. The numbers given in line 383 are without any error estimates.
15.	Figure 9,10. Emergence decreases with training – so it is negatively correlated with performance. But it’s higher for the pre-trained networks – so it is positively correlated with performance.
16.	Line 466 further describes “The direct relationship between the change in training accuracy and the emergence value validates the the value being a predictor of the networks performance.”. But the blue lower curve jumps in emergence at epoch 17 while the change in performance is around epoch 25. The purple curve (which is not color matched) has a non-trivial performance curve, but nothing corresponding in emergence.

---

### Official Review · Reviewer_825o · 2024-11-03

**Soundness:** 1
**Presentation:** 1
**Contribution:** 2
**Rating:** 3
**Confidence:** 3

**Summary:**

The authors tackle the concept of emergence, and specifically its relationship with training performance. The authors build recent work to define their chosen metric of emergence. The relationship with training is first studied empirically in experiments on (fashion)-MNIST and CIFAR-10. In parallel, the authors explore how the emergence relates to the loss landscape, and probe the concept of tuning under this light. Finally, the authors open up the discussion on how such an emergence metric can help understand training, pruning, and provide a tool for training convergence and consequently early training-termination.

**Strengths:**

- The paper is original, approaching the relevant question of quantifying the wide-spread phenomenon of emergence, which lacked rigorous treatment in the past.
- The paper is well motivated. L35 for instance is a perfect example of a well-posed objective.
- The paper builds on previous work on defining emergence, and applies it to the extremely well-studied setting of training in NNs performing classification tasks, thus making the results approachable to a wide audience.
- The authors identify some covariates that could bias their estimates and correct for it, such as the number of parameters changing during pruning.
- Some results are quite clear. For instance, relative emergence being correlated with pruning (Figure 4 and 8) is well supported.
- The implications of an emergence metric for training termination are interesting.

**Weaknesses:**

Major:

- Conclusions are made before results are presented (L62, L256, L266), and sometimes some claims are made with no supporting results at all (L242, L302). Furthermore, many results are presented as "validating our hypothesis". I would suggest removing this hypothesis-based presentation, and focus on an analysis of your results. This could be done by presenting the results of section 4 first, *then* dive into the analyses of the latter half of section 3.3(.0) and section 3.3.1.

- Mathematical definitions lack support and introduction. The definitions rely on (Li et al, 2023), which "provides rigorous mathematical foundation for understanding emergence in network structures" (L77). The authors would benefit from integrating in their manuscript a more thorough summary and introduction to the definitions. For instance, it is highly unclear to me how one goes from equation on L166 to the main definition of emergence in L175, and how the latter can be derived from Eq. (8) in Appendix. Appendix A does not provide clarification unfortunately, only copy-pasting (exact replica with unproperly formatted references, see L648 and L669) from (Li et al., 2024).

- Lack of surrounding literature. At a superficial level, the reference list is quite small, with more than half of the papers being preprints/non-peer-reviewed work. More concretely, the authors would benefit from including more literature on emergence in LLMs and neural scaling laws (see e.g. (Nam et al., 2024, NeurIPS) for recently published work that surveyed some of these topics).

- Circular reasoning arguments are employed many times, starting with the abstract "Our hypothesis posits that the degree of emergence [...] can predict the development of emergent behaviors in the network", then further in L381 "as the value of Emergence decreased, training accuracy change decrease, thereby supporting our hypothesis that emergence functions as a measure of prediction of emergent traits within a neural network.", and again on L438 "[...], supporting the idea that the emergence value is a predictor of emergent traits." . These are logical fallacies unfortunately and should be removed, in a favor of exposition in line with my first point above.

- Repeated sentences and content, for instance L44-L47 and Paragraph beginning with L71 both makes similar comments on the loss landscape and its relationship with emergence, and see a further example in the circular reasoning point above.

- Claims of significance should be backed by statistical tests.

Minor
- Repeated reference for (Li et al, 2023)
- Remove "intuitively" in L251 if no intuition is provided.
- Many typos, too many to list unfortunately. A thorough re-read is advised.

**Questions:**

- What are baselines for your metric of emergence?
- How is the threshold of 0.05 chosen? How do your results vary as a function of that threshold? Weight values are related to performance in a very complex, non-trivial manner (for instance as highlighted in the rich v.s. lazy literature on learning dynamics), and while you do touch this aspect in section 4.2., the paper would benefit from more justification of the chosen bounds generally.
- To provide some intuition, how does the metric behave under alternative transformations $\Phi$ not representative of training, for instance, additive Gaussian noise on the weights? Why does it decrease over training? (Figure 4, 8)
- What is Figure 2 showing? If it corresponds to scalar E over a parameter state space, what are these parameters, and how are the (un)pruned networks plotted against it?
- I think the results on emergence and end of training are interesting: could you explore further the possible advantages, for instance by comparing how many epochs earlier you can predict the end of training than traditional measures of convergence?

---

### Official Review · Reviewer_Qumd · 2024-11-03

**Soundness:** 2
**Presentation:** 2
**Contribution:** 1
**Rating:** 3
**Confidence:** 3

**Summary:**

In this manuscript, the authors analyzed the learning dynamics of deep neural networks in image recognition tasks from the perspective of “emergence.” They defined “emergence” as the total number of pathways from an inactive node to active node in the neural network. The authors numerically compared “emergence” with training performance across various image recognition tasks, concluding that the “emergence” measure correlates with performance improvement.

**Strengths:**

The manuscript is clearly written. The emergence of functionality that the authors aimed to address is an important topic in deep learning that we have yet to understand well.

**Weaknesses:**

I'm afraid that this manuscript falls below the standard in several aspects.

The definition of “emergence”: The definition of “emergence” does not seem to capture the emergent phenomena discussed in the introduction and related work. The authors define “emergence” based on the number of active units in each layer, effectively making it a measure of hidden layer activity sparsity. Specifically, according to the definition provided in lines 200–201, maximizing "emergence" would require all units in the first layer to be inactive ($a_1=0$) and all units in the last layer to be active ($a_N = n_N$). It suggests that the measure may not be meaningful in the context of emergent phenomena.

Relationship between the loss landscape and the “emergence” measure: The discussion of this relationship lacks substance. The authors included only a schematic figure without providing analytical or numerical support.

Impact of pruning on performance: It is well-established that neural networks can retain learning performance even with up to 99% pruning (e.g., Lee, Ajanthan & Torr, ICLR 2019). Thus, the observed high performance in a model pruned by 70% is not surprising.

Correlation between “emergence” and task performance: The presented figures do not clearly show a correlation between “emergence” and task performance, except in the 0% pruning scenario. Additionally, it is unclear what insight this correlation offers, given that activity sparsity naturally changes during training.

Lack of essential details in numerical experiments: Key details are missing. For instance, how is a hidden unit determined to be active? Does a unit need to exceed an activity threshold of 0.05 across all samples, or is it sufficient for the average activity over samples to exceed 0.05? What learning algorithm was used for training? What are the width and depth of the network?

**Questions:**

Could you clarify the key details of the numerical experiments, especially the points above?

---

### Official Review · Reviewer_YP1b · 2024-11-08

**Soundness:** 1
**Presentation:** 2
**Contribution:** 1
**Rating:** 3
**Confidence:** 3

**Summary:**

The authors use the framework of emergence as defined in Li et al 2023a to experimentally study the role of emergence during training. Emergence is varied by pruning the network or initializing it with high vs low emergence. Then variations in the training curves of these networks is attempted to be explained by absolute or relative emergence.

**Strengths:**

The authors attempt to show practical uses of Emergence as defined in Li et al 2023a, Li et al 2024 to explain the loss landscape and training curves in neural networks.

**Weaknesses:**

Since no new theory is developed, the authors should at least compare this framework to others that explain training behaviour, e.g. double descent or the lottery ticket hypothesis (see e.g. proofs in Malach et al 2020), if they want to position Emergence as a useful metric.

Figure 2 seems just an illustration of what the authors imagine the energy landscape would look like. Why "relative emergence can be conceptualized as reflecting the density of local minima within a given region of the loss landscape" is not substantiated by the authors.

Figures 3-8 appear to have been run without multiple seeds as I see no standard deviations around the plots? Various arguments are made about minor differences in training accuracy and emergence evolution when all these seem within the margin of noise around these curves.

**Questions:**

All the weakness above need to be addressed satisfactorily. I think this would require a lot more simulations and ( experimental or theoretical) justifications of the claims, along with comparisons to other frameworks that explain learning behavior of neural networks.

Line 339: Rather than maintaining the learning rate constant, might be better to tune the learning rate as a hyperparameter across also 4 models. Because after pruning, depending on LayerNorm or otherwise, the fraction of total input to a neuron due any one parameter changes, and so the max learning rate without losing stability will also change and so learning rate should be tuned for each model separately. Or just use Adam.

Minor:
line 319: paramters -> parameters
Figure 3/4: Please use colors that are not so similar. Like red, blue, etc. Currently all 4 lines are different shades of red, which are very difficult to distinguish.

---

### Meta-Review · Area_Chair_nzh6 · 2024-12-10

**Metareview:**

This paper proposes a measure of "emergence" in neural networks that uses the count of the number of paths from inactive nodes
to active nodes. The authors claim that their measure can predict the appearance of "emergent behaviours", i.e. new capabilities appearing in the network. The also claim that emergence, per their definition, relates to the smoothness of the loss landscape and they claim that the framework can be used to understand the impact of pruning on performance.

The strength of this paper is that it is addressing an interesting question that many researchers would like to understand better. There are numerous weaknesses to the paper though. Most notably, the definition of emergence, as provided, leads to some trivially uninteresting methods for increasing "emergence" (such as having sparse activity in early layers and dense activity in late layers). As well, some of the logic is circular, basically using the definition itself to provide evidence of the validity of the idea. There are also numerous technical problems, such as a lack of multiple seeds or insufficient details on experimental parameters. Given these considerations, a decision of reject was natural.

**Additional Comments On Reviewer Discussion:**

The reviewers provided a robust set of critiques of the paper, but the authors did not respond. As such, none of the points were addressed by the authors.

---

### Decision · Program_Chairs · 2025-01-22

Reject